

# Global opportunities for mariculture development to promote human nutrition

Owen R. Liu[1], Renato Molina[1,2], Margaret Wilson[1] and Benjamin S. Halpern[1,3,4]

[1] Bren School of Environmental Science and Management, University of California, Santa Barbara, Santa Barbara, CA, United States of America
[2] Department of Economics, University of California, Santa Barbara, Santa Barbara, CA, United States of America
[3] National Center for Ecological Analysis and Synthesis, Santa Barbara, CA, United States of America
[4] Imperial College London, Ascot, United Kingdom

## ABSTRACT

An estimated two billion people worldwide currently suffer from micronutrient malnutrition, and almost one billion are calorie deficient. Providing adequate nutrition is a growing global challenge. Seafood is one of the most important sources of both protein and micronutrients for many, yet production from wild capture fisheries has stagnated. In contrast, aquaculture is the world's fastest-growing food production sector and now supplies over half of all seafood consumed globally. Mariculture, or the farming of brackish and marine species, accounts for roughly one-third of all aquaculture production and has received increasing attention as a potential supplement for wild-caught marine fisheries. By analyzing global patterns in seafood reliance, malnutrition levels, and economic opportunity, this study identifies where mariculture has the greatest potential to improve human nutrition. We calculate a mariculture opportunity index for 117 coastal nations by drawing on a diverse set of seafood production, trade, consumption, and nutrition data. Seventeen primary variables are combined into country-level scores for reliance on seafood, opportunity for nutritional improvement, and opportunity for economic development of mariculture. The final mariculture opportunity score identifies countries with high seafood reliance combined with high nutritional and economic opportunity scores. We find that island nations in Southeast Asia and the Caribbean are consistently identified as countries with high mariculture opportunity. In other regions, nutritional and economic opportunity scores are not significantly correlated, and we discuss the implications of this finding for crafting appropriate development policy. Finally, we identify key challenges to ameliorating malnutrition through mariculture development, including insufficient policy infrastructure, government instability, and ensuring local consumption of farmed fish. Our analysis is an important step towards prioritizing nations where the economic and nutritional benefits of expanding mariculture may be jointly captured.

Corresponding author
Owen R. Liu, oliu@bren.ucsb.edu

## INTRODUCTION

With large uncertainty surrounding the future of wild caught fisheries, the potential role of farmed fish has gained increasing attention in global nutrition conversations (*Beveridge et al., 2013*; *Béné et al., 2015*; *Golden et al., 2016*; *Little, Newton & Beveridge, 2016*). An estimated one billion people are calorie deficient, and two billion suffer from micronutrient malnutrition (*IFPRI, 2016*). Zinc deficiency affects 17% of the global population (*Golden et al., 2016*) and is responsible for an estimated 800,000 annual child mortalities (*FAO, 2016*). Nearly one-third of the world's population is iron deficient (*FAO, 2016*) and one-fifth of maternal deaths are linked to anemia during pregnancy (*Micronutrient Initiative, 2009*). Vitamin A deficiency is the leading cause of preventable blindness and affects an estimated 250–500 million children, half of whom will die within a year of vision loss (*Bailey, West & Black, 2015*).

Seafood is a critical source of all of these nutrients. Fish currently provides 17% of the world's animal protein, and exceeds 50% in the diets of many least-developed countries (*FAO, 2016*). One of the most documented nutritional benefits of seafood is the linkage between complex fatty acids found in fish and their contribution to brain development, metabolic function, and the prevention of cardiovascular disease (*Larsen, Eilertsen & Elvevoll, 2011*). But seafood in general also provides essential micronutrients that promote healthy growth and development, particularly in children and pregnant women (*Kawarazuka & Béné, 2011*; *Béné et al., 2015*; *FAO, 2016*). Nevertheless, declines in global wild fish stocks paired with a predicted human population of nearly 10 billion by 2050 may leave even greater numbers at risk of nutrient deficiency (*UNDP, 2015*; *Blasiak et al., 2017*). *Golden et al. (2016)* estimate that an additional 11% of the population is vulnerable to zinc, iron, and vitamin A deficiencies as fish stocks decline in coming decades, and nearly 20% for all micronutrients exclusive to animal food sources, such as fatty acids and vitamin B12.

Due in part to the nutritional importance of fish and significant technological advances to produce seafood (*Kumar & Engle, 2016*), its consumption has more than doubled from 9.9 kg per capita in the 1960s to a current average of 20.2 kg (*FAO, 2016*). Global fish consumption is predicted to increase more than 20% by 2025, as both human population and economic development rise in coming decades (*FAO, 2016*). Driven by this increasing demand, aquaculture has been the fastest growing food production sector for four decades and now exceeds wild fisheries production (*Tveteras et al., 2012*; *Troell et al., 2014*). About one-third of this total production comes from the farming of marine species, also known as mariculture (*Ottinger, Clauss & Kuenzer, 2016*). While issues around freshwater scarcity (*Verdegem & Bosma, 2009*) and pollution (*Cao et al., 2007*; *Edwards, 2015*) may slow the growth of freshwater aquaculture in coming years, mariculture has been identified as an area of high growth opportunity (*Holmer, 2010*; *Kapetsky, Aguilar-Manjarrez & Jenness, 2013*; *Gentry et al., 2017*).

Increased mariculture production could help ameliorate global malnutrition, but its current development typically excludes lower-income countries or is marketed towards trade with wealthier countries and consumers (*Watson et al., 2015*; *Asche et al., 2015a*;

*Golden et al., 2016*; *Golden et al., 2017*). Global mariculture production currently focuses predominantly on high-value species like salmon, shrimp and tuna, which largely go to global markets (*Bostock et al., 2010*). It remains unknown whether mariculture can meaningfully contribute to global nutrition, in part because no previous analysis has identified countries where economic and nutritional development opportunities are expected to overlap. Before developing any strategies to link these objectives, however, it is critical to first identify key overlaps between nutritional needs and economic opportunity for further mariculture development.

Here we provide global analyses to identify countries where joint economic and nutritional mariculture development may be most synergistic. Our motivating question is, where do nutritional needs—needs that can be effectively alleviated by seafood consumption—overlap with economic development opportunities for mariculture? By using global datasets and developing a comparative scoring system, we identify high-opportunity countries via an analysis of country-level malnutrition, seafood reliance, and economic opportunity. We dissect emergent patterns in the global analysis and discuss their potential drivers. Finally, we identify common development obstacles that may be applicable to future global mariculture ventures.

## METHODS

### Defining mariculture opportunity

For a country to tackle the nutritional deficiencies of its population through mariculture development, it should have three main characteristics, expressed herein through three scores that we compile for each nation in our analysis. First, the country should have a demonstrated need for the macro- and micronutrients that seafood can provide. As described above, seafood can be an efficient and important source of not just calories, but also protein, healthy fatty acids, zinc, vitamin A, and iron (*Kawarazuka & Béné, 2011*; *Béné et al., 2015*). On the other hand, countries that are well-nourished will not necessarily benefit (nutritionally) from adding more fish to the diet. We refer to a country's relative deficiencies in these key nutrients as the country's nutritional opportunity.

There should also be good evidence within the country of a cultural predisposition to seafood consumption. Clearly, increases in mariculture production will be most directly important for alleviating nutritional deficiencies if seafood accounts for a large proportion of a country's diet. For this reason, we also include seafood reliance—calculated as the relative contribution of seafood to total diet—as a core enabling factor for mariculture opportunity.

Finally, a country's mariculture production should be economically viable in order to sustainably provide a nutritional solution. Many combined mariculture/development projects fail to be sustainable because of a lack of scalability or long-term economic feasibility (*Béné et al., 2016*; *Little, Newton & Beveridge, 2016*). Hence, our third score for each country is a measure of this economic opportunity, constructed from each country's current aquaculture production and seafood trade data, as well as proxies for the value of the seafood production sector and latent economic development potential.
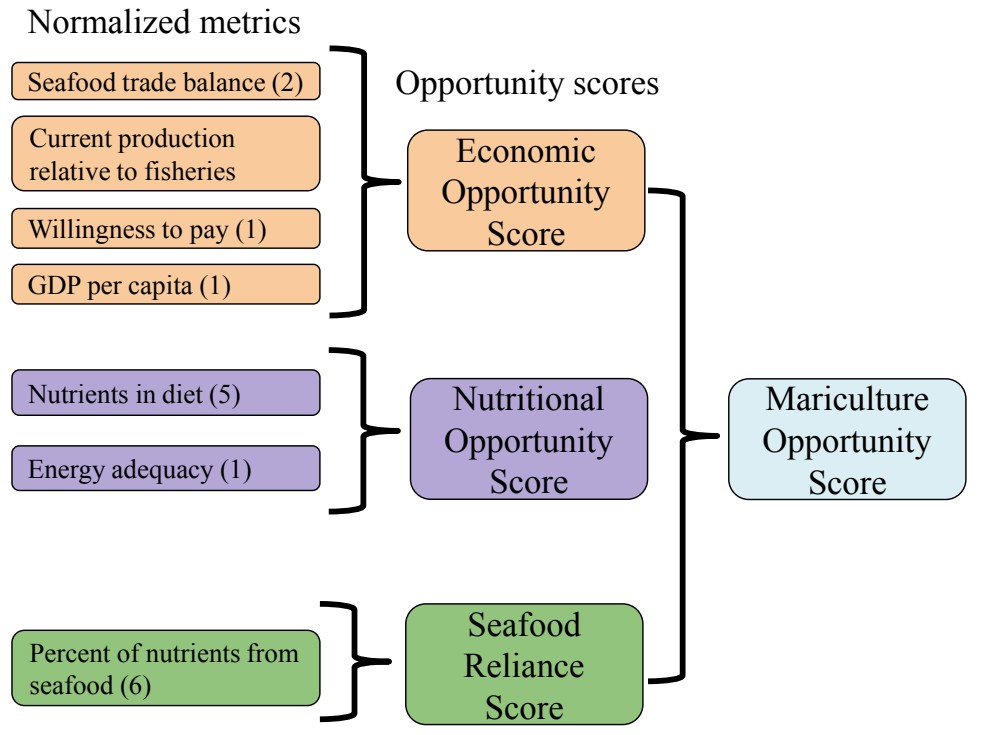

**Figure 1** **Schematic of mariculture opportunity score calculation for each nation.** Categories of metrics consolidated for clarity, with total number of raw variables in parentheses. See Eqs. (2)–(5) for score calculation.

## Mariculture opportunity metrics

We compiled raw data for economic opportunity, nutritional opportunity, and seafood reliance from two publicly available databases (Fig. 1). We endeavored to limit our metrics to those that are directly relevant to the economic development of mariculture and the alleviation of nutritional deficiencies through seafood production. The resulting set of raw data includes five economic opportunity metrics, six nutritional opportunity metrics, and six seafood reliance metrics by country. To facilitate global comparisons, these 17 raw metrics were normalized and then combined into the three opportunity metrics and a final mariculture opportunity metric (Fig. 1).

Nutritional opportunity and seafood reliance scores were calculated using raw metrics from the Harvard GENuS database (*Smith et al., 2016*, https://dataverse.harvard.edu/dataverse/GENuS). GENuS models comprehensive country-specific diet and nutrient supply information by extrapolating from the Food and Agriculture Organization of the United Nations' (FAO) food balance sheets, household surveys, and production data. GENuS estimates per capita nutrient consumption across hundreds of food categories. For the purposes of this study, we utilized data on the average daily per capita intake by country of five essential nutrients that can be obtained from seafood: protein, vitamin A, zinc, iron, and polyunsaturated fatty acids (PUFAs). Separately, we also collected each country's average Dietary Energy Supply Adequacy, an FAO measure of the basic adequacy of total

caloric intake relative to a sufficient diet (Eqs. (1) and (2)). These data—energy adequacy plus the daily per capita intake of five nutrients—comprise our six nutritional opportunity metrics. Together, these measures provide a synthesis of the average nutritional status of each country, specific to those nutrients that mariculture products can provide.

Our raw seafood reliance data were also drawn from the GENuS database. Because GENuS provides per capita nutrient intakes by food-group, we were able to sum per capita intakes from all FAO marine harvest categories (pelagic fish, demersal fish, other marine fish, crustaceans, and mollusks) to calculate total nutrient and calorie intakes obtained from seafood. We divided these seafood-specific intake values by total per capita intake values to calculate the percent of each nutrient obtained from seafood products. Six of these percentage values—for calories, protein, vitamin A, zinc, iron, and PUFAs—comprise our six seafood reliance metrics. Having both average nutritional status (nutritional opportunity score) and seafood reliance allows our scoring system to identify countries where increased mariculture production may have the greatest chance to directly address nutritional deficiencies, and where vulnerability to potential declines in wild-caught fisheries is highest.

The third dimension of mariculture opportunity is economic opportunity. Economic metrics were drawn from FishStatJ (http://www.fao.org/fishery/statistics/software/fishstatj/en), a freely available software used to access data from the Fisheries and Aquaculture division of FAO. FishStatJ provides panel data on fisheries and aquaculture production and trade by country, species, and commodity type. Selecting only the most recent year for which all metrics are available (2011), and excluding all commodity categories not for direct human consumption (e.g., fish meal or fish oil), these data were analyzed to produce the five economic opportunity metrics for each country: (1) production ratio, (2) trade balance in terms of quantity, (3) trade balance in terms of value, (4) GDP per capita, and (5) willingness to pay for seafood.

We define a country's production ratio as its total aquaculture production divided by its total marine fisheries production in metric tons. Both production metrics were drawn directly from FAO reported data. This measure serves as a proxy for relative importance of two sectors that share infrastructure and markets. The logic is that countries with active fishing sectors should have both capital and management institutions that could also be functional to production and regulation of mariculture. The balance of fisheries and aquaculture production determines the opportunity for mariculture to utilize that shared infrastructure. The more skewed the production ratio is toward fisheries, the more potential there is to take advantage of these overlaps through the further development of a mariculture sector. While an indirect proxy for infrastructure, production ratio was chosen because of its generality across multiple types of potential mariculture production and its consistency across countries.

Two of our economic opportunity metrics measure trade balance in quantity and value. In our study, trade balance describes each country's total volume or value of exports of seafood products (not just mariculture) divided by its imports. Trade balance measured in this way is a proxy for how a country balances supply and demand in the global seafood market. Trade imbalances reveal how countries compensate for their domestic seafood

demand: a trade imbalance in which imports outweigh exports implies an opportunity to satisfy excess demand with augmented domestic mariculture production. Because seafood products vary so widely in their value relative to their volume, this trade balance signal could manifest in either metric, hence our inclusion of both quantity and value metrics.

The metric for GDP per capita is included as a proxy for latent economic opportunity. Countries with low per capita GDP have a need for economic development that may be partially pursued through mariculture. In this way, lower GDP per capita corresponds to higher economic opportunity scores in our analysis.

Willingness to pay for seafood, our final economic opportunity metric, is defined as a country's total value of seafood imports divided by its GDP. This measure serves as a proxy for seafood value in each country. Although an imperfect metric, as it combines high-volume, low-value seafood with high-value niche products, willingness to pay still reflects overall expenditure on foreign seafood production. In our analysis, a higher willingness to pay corresponds to a greater opportunity to capture that willingness to pay with mariculture products produced domestically. In the calculation of opportunity scores in the next section, we use the reciprocal value of willingness to pay so that its ordering aligns with the other economic metrics (a lower value of the metric corresponds to a higher economic opportunity).

Together, these economic measures provide the essential information to describe a given country's current mariculture production status relative to other nations. Furthermore, by utilizing FAO data, this set of economic metrics provides the ability to contrast countries while reducing potential sources of inconsistency and bias that might arise from using disparate sources, while at the same time being readily amenable to update as new data become available. Each individual metric provides one perspective on the enabling conditions for economic development of mariculture. Based on our economic opportunity metrics, a country with a high economic opportunity is one with existing seafood industry infrastructure, a seafood trade balance that could benefit from increased domestic production, a demonstrated value of seafood in the country, and a relatively low per capita GDP.

## Calculation of mariculture opportunity scores

The 17 metrics were combined into three opportunity scores and one final mariculture opportunity score (Fig. 1). All raw metrics were normalized to a zero to one scale to allow comparison across categories of metrics. For each metric, $X_i$ was scaled by dividing by its 80th percentile value across countries (Eq. (1)).

$$X_i = \frac{X_{raw,i}}{P_{80}[X_{raw}]}. \tag{1}$$

This scaling was chosen to reduce the influence of large single-metric outliers on the ability to distinguish between nations. The choice had little effect on the final ranks of nations compared to dividing by the 90th percentile or simply the maximum value for each metric (see Table A2; final country ranks between three alternate scaling choices were significantly concordant; Kendall's $W = 0.925$, $p < 0.05$).

The three scores–nutritional opportunity, seafood reliance, and economic opportunity–for each nation were determined by calculating a mean across the set of normalized metrics associated with that score (Eqs. (2)–(4), Fig. 1). No weighting was done in the calculation of aggregated scores because our emphasis is on countries' relative positions. There was no definitive rationale for weighting any metric more heavily than any other, and doing so might unnecessarily complicate the interpretation of our scoring system and results. While we did not choose to use weighted scores, our methodology remains flexible to that extension.

Equations (2)–(5) describe score calculation. Country $i$'s economic opportunity score (Eq. (2)) was defined as the mean of its normalized metrics for production ratio, trade balance in value, trade balance in quantity, willingness to pay, and GDP per capita. The country's nutritional opportunity score is the mean of its normalized intakes of protein, vitamin A, zinc, iron, and PUFAs, as well as its normalized FAO energy adequacy. Because the raw dietary nutritional supplies do not scale to zero–no country's diet consists of zero calories–the set of nutritional opportunity scores were further rescaled by subtracting the minimum country score and dividing by the range across all nutritional opportunity scores. Finally, a country's seafood reliance score is the mean of its normalized metrics for protein, vitamin A, zinc, iron, fatty acids, and calories derived from seafood. Each of the three metrics was ordered such that a higher score (closer to 1) corresponds to higher opportunity in that dimension, as described in the previous section.

$$Econ_i = 1 - Average(ProdRatio_i + TradeQ_i + TradeV_i + \frac{1}{WTP^i} + GDPpc_i). \qquad (2)$$

$$Nutri_i = 1 - Average(Protein_i + VitA_i + Zinc_i + Iron_i + PUFA_i + Energy_i). \qquad (3)$$

$$Reliance_i = Average(ProtSea_i + VitASea_i + ZincSea_i + IronSea_i + PUFASea_i + CalSea_i). \qquad (4)$$

$$Opportunity_i = Average(Econ_i + Nutri_i + Reliance_i). \qquad (5)$$

A final mariculture opportunity score for each country was calculated by averaging its economic opportunity, nutrition opportunity, and seafood reliance scores (Eq. (5)).

Non-coastal nations ($N = 49$) were excluded from the analysis before score calculation, because we were focused on the potential for local mariculture to address in-country nutritional needs. We likewise removed countries missing entire categories of data (e.g., missing all nutritional data). To avoid overly biasing our sample against data-limited countries, we retained countries with missing aquaculture production ($N = 23$), dietary energy supply adequacy ($N = 10$), and GDP ($N = 4$) values and simply omitted those individual variables from the countries' score calculations. Sensitivity analyses on nations with complete data revealed that the effect on countries' opportunity scores of single missing metrics was minimal, so gap-filling procedures (and their associated uncertainty) were not deemed necessary (Table A3). The final sample consists of 117 coastal nations. All raw and normalized metrics and scores for each country are available in the Supplemental Information.

## Limitations of the analysis

The metrics described in the section above have two main advantages: (i) the data required is readily available through standard and credible data sources, and (ii) the calculation of

each metric is straightforward and amenable to further refinement. However, the metrics also have two important limitations that warrant mention, and both are fully explored in the Appendix.

First, our metrics are constructed with cross-sectional data. This feature impedes our ability to explicitly examine past trends or predict future trajectories. But any cross-sectional approach suffers from the same limitations, and we are forced to rely on the assumption that the data reflects the actual realities of every country for that specific point in time.

Particular to our case, the limiting factor is that data for nutritional intakes (GeNUS) are only available for 2011. If a panel dataset on nutritional intake were to be available, it may prove beneficial to work with trends by country, rather than single observations for specific years. Unfortunately, to the best of our knowledge, these data are unavailable. We therefore use cross-sectional (2011) data in our main analysis, but in the Appendix we provide an extension to test the sensitivity of our results to the economic data, which are available in time series.

Secondly, it is possible that our seafood reliance score reflects preferences rather than pure reliance. For example, rich countries showing high seafood reliance could be merely a result of the average consumer's preferences and may have little to do with the overall seafood availability or its price. If the index reflects preferences rather than pure reliance, the interpretation of the seafood reliance may be altered. This is an important issue if we are trying to establish the potential of mariculture in solving nutritional needs, but at the same time it is intrinsic to how these aggregate measures are collected across nations. We address this limitation in the Appendix by extending our main analysis and sorting countries in the sample based on their income level as reported by the World Bank (Fig. A3), as well as their geographical characteristics (Continental or Island; Fig. A2). Breaking down our results in this manner allows us identify the relative influence of income and country type on seafood reliance.

## RESULTS

Final mariculture opportunity scores are mapped in Fig. 2. Countries with high mariculture opportunity scores (orange and red) are places where mariculture has the highest potential to ameliorate nutritional deficiencies because of the apparent alignment between nutritional and economic opportunities and a demonstrated reliance on seafood. These multi-dimensional opportunities are apparent for some nations, especially island nations in the Caribbean (Fig. 2B) and Southeast Asia. Other nations, notably in parts of Europe and Africa, also had high mariculture opportunity scores.

In addition to geographic patterns in final mariculture scores, several patterns emerged for each of the three separate opportunity scores (Fig. 3). First, countries' economic opportunity scores were generally clustered towards greater opportunity (mean score 0.59 +/− 0.2, Fig. 3). This pattern indicates that comparatively few nations have developed mariculture industries, while the bulk of nations have potential to further develop mariculture. Indeed, the few top producers received low economic opportunity scores. China, Indonesia, the Philippines, and Norway, which are four of the five top

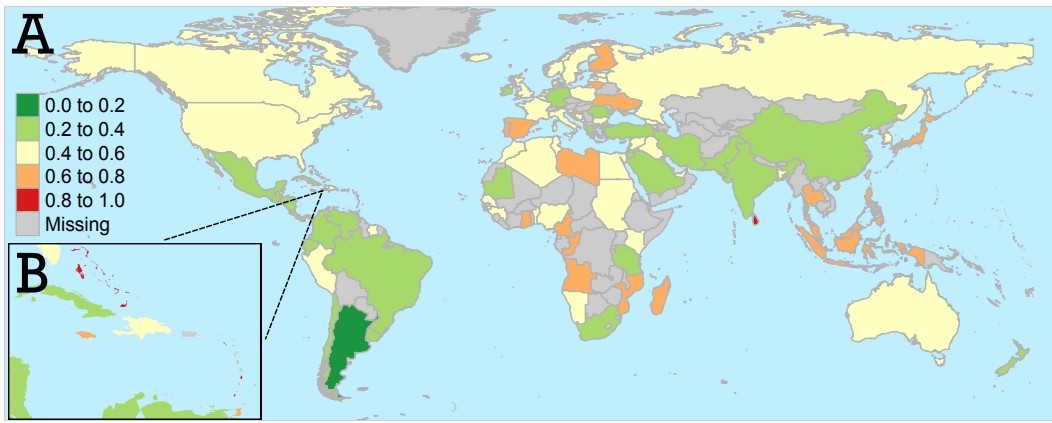

**Figure 2** (A) **Final mariculture opportunity scores for the entire world, with (B) detail for the Caribbean region.** Gray indicates countries which were removed from the analysis (see 'Methods') or no data were available. World borders dataset from thematicmapping.org, obtained under a Creative Commons license.

current producers of mariculture (*FAO, 2016*), all received scores less than 0.35. Overall, mariculture production level was significantly negatively correlated with economic opportunity (Pearson's $r = -0.22$, $p < 0.05$).

In contrast, relative to the economic scores, nutritional opportunity scores were generally lower, or right-skewed (mean nutritional opportunity score $0.42 +/- 0.23$, Fig. 3). Europe and Asia were the regions with the overall lowest nutritional opportunities, while the Southeast Asia/Oceania and Latin America/Caribbean regions show the highest nutritional opportunity.

This combination of higher economic opportunity and lower nutritional opportunity creates the cluster of nations in the lower right quadrant of Fig. 3. Fifty-three of 117 nations, or 45%, have an economic opportunity score greater than 0.5 and a nutritional opportunity score less than 0.5. Overall, economic and nutritional opportunity scores are positively correlated, but the correlation is not significant (Pearson's $r = 0.15$, $p = 0.1$).

Nutritional opportunity and seafood reliance show significant positive correlation overall (Pearson's $r = 0.23$, $p = 0.01$). Nine of the top 10 country scores (and 15 of the top 20) for seafood reliance come from island nations. Countries with a higher reliance on seafood generally have higher nutritional opportunity scores, although there were differences in the relationship between reliance and nutrition between geographic regions. Southeast Asian countries (purple dots in Fig. 3) have generally high nutritional opportunity and high seafood reliance, but scatter along a spectrum of economic opportunity. In contrast, Latin American and Caribbean countries (teal dots) display a high correlation between nutritional opportunity and seafood reliance: low nutritional opportunity corresponds with low reliance, and vice versa. European countries display a range of seafood reliance, but generally low nutritional opportunity.
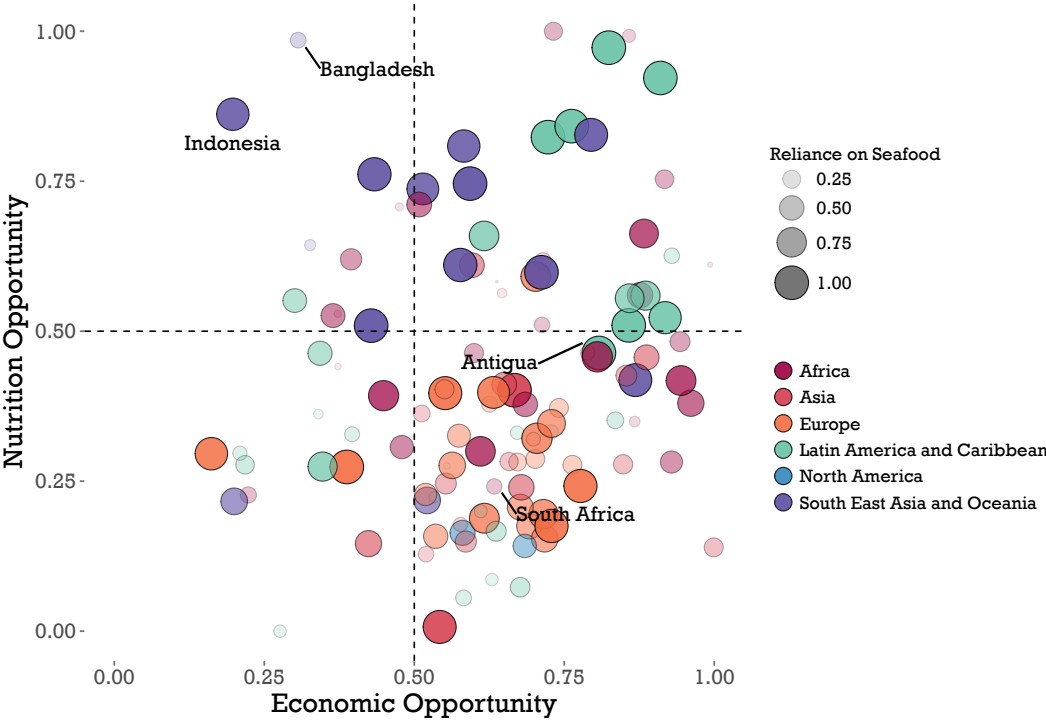

**Figure 3** **Results of global analysis of nutritional opportunity, seafood reliance, and economic opportunity.** Individual countries ($n = 117$) scatter along economic and nutritional opportunity scores on the $x$ and $y$ axes, respectively, where each point indicates the performance of a given country. Scores are scaled from zero to one (see 'Methods'), such that countries in the upper right quadrant have both a high economic and nutritional opportunity for mariculture development. Size and opacity of country points scale with each country's seafood reliance score, while color indicates a country's geographic region. Countries referred to in the Discussion are labeled.

## DISCUSSION

The objectives of our analysis were to identify opportunities for nutritional improvement through mariculture and to use multidimensional opportunity scores to inform future development efforts. Our results identify countries with poor nutrition and high reliance on seafood. Of those countries, our overall mariculture opportunity score prioritizes those with simultaneous large nutritional and economic opportunities for mariculture development. Other analyses have projected large increases in production in many of the nations—especially in Latin America, the Caribbean, and Southeast Asia—which we identified as having high nutritional opportunity (*Kobayashi et al., 2015*). Mariculture has the potential to benefit malnourished populations in these countries both directly through increasing seafood availability and indirectly through economic gains (*Béné et al., 2016*), and our analysis clearly identifies places where these opportunities exist. But despite this theoretical potential, limited evidence exists suggesting mariculture will address local nutritional needs in reality (*Beveridge et al., 2013*; *Béné et al., 2016*; *Golden et al., 2016*; *Golden et al., 2017*; *Little, Newton & Beveridge, 2016*). What barriers are preventing the
potential nutritional benefits of mariculture from being realized, and how can we use this global analysis to guide nutritionally focused development strategies?

The link between mariculture, or aquaculture in general, and the amelioration of malnutrition has not proven inherent (*Golden et al., 2017*). Numerous countries have already developed fish farming industries but still struggle with malnutrition. Our analysis corroborates this disconnect by finding many nations with a low economic opportunity but high seafood reliance and nutritional opportunity (upper left quadrant in Fig. 3). In these scenarios, aquaculture production is not being translated into nutritional gains for at-risk populations. This widespread disconnect between aquaculture production and local nutrition is largely a reflection of the industry's historical development. Private investment opportunities—as opposed to nutritional necessity—have been the primary drivers of aquaculture growth, especially in recent decades (*Little, Newton & Beveridge, 2016*). As industries have developed, improved productivity of larger farms and increased international trade have incentivized consolidation and export-oriented operations in many producing countries (*Asche, Roheim & Smith, 2015b*; *Little, Newton & Beveridge, 2016*).

This trend, however, does not necessarily mean that growth in mariculture production is or will be completely at the expense of the malnourished poor. *Toufique & Belton (2014)* found a convincing positive link between large-scale growth of pond aquaculture in Bangladesh and fish consumption by the extreme poor. Although the debate surrounding the strength of the link between mariculture development and nutrition improvement remains active, it is clear that the two are not necessarily mutually exclusive. Our analysis provides a guide to the countries where this link might be more effectively forged.

In this context, opportunity costs may pose a major barrier to nutritionally-focused mariculture development. Small island nations were overwhelmingly identified as high-opportunity countries in our analysis. Many of these nations, especially in the Caribbean, are on a development path focused on the promotion of tourism and importation of wealth from abroad (*Ashe, 2005*; *Scheyvens & Momsen, 2008*). Because of the tourism industry, coastal property is at a premium, turning coastal or near-shore mariculture activities into a non-competitive investment alternative. A shortage in affordable coastal real estate might incentivize development of offshore mariculture, though these systems will require significant amounts of external investment and technological capacity. To stay profitable, high cost systems will likely focus on high market value species intended for export and/or consumption by higher income individuals. Several mariculture initiatives in the Caribbean have already been designed in this manner. In Antigua, for example, a private mariculture initiative led by an American and European board of directors plans to develop "high-tech" offshore pens to raise Kampachi (*Seriola rivoliana*), a high value, sashimi grade fish for export (http://www.asacip.com/). While these projects typically promise to provide local employment, the expected outcome for local nutrition remains unclear (*Béné et al., 2016*). Hence, while our analysis may identify island nations as an important focus for mariculture development and nutrition, capturing this overlapping opportunity will be a policy challenge unique to each country or project.

In cases where mariculture products are made locally accessible, effectively addressing malnutrition issues also requires significant education and marketing programs at local to

regional scales. Our opportunity criteria prioritized countries with high existing seafood reliance. Existing culture around eating seafood in these countries may facilitate a transition to consuming mariculture products, though local attitudes towards farmed fish may prove a significant barrier. Case studies in Bangladesh reveal that farmed fish are typically harvested at a larger size and consumed filleted, which may provide less nutritional value than the small indigenous fish that are traditionally consumed whole (*Kawarazuka & Béné, 2011*). Nutrition programs can play a critical role in educating the public on product selection and preparation in order to maximize nutritional effectiveness.

While our chosen scoring system prioritizes countries with high seafood reliance, there are certainly opportunities to link mariculture production with local nutritional benefit when reliance is low, and dietary contribution of seafood may increase with development (see Appendix). These efforts, however, will require even greater investment in social planning and policymaking to ensure these products are reaching nutritionally vulnerable populations. Alternatively, the direct involvement of poor sectors in the mariculture industry could increase disposable income and, consequently, access to nutritious food. This was demonstrated for aquaculture in Malawi (*Aiga et al., 2009*), though these types of indirect benefits require further investigation (*Béné et al., 2016*; *Golden et al., 2017*).

A further challenge in mariculture development is to mitigate environmental harm to the extent possible. A recent study establishes that there are vast areas suitable for development of mariculture in almost every coastal nation (*Gentry et al., 2017*). Nonetheless, while suitable space is likely not limited, intensive mariculture development comes with a host of potential environmental problems, including pollution, habitat destruction, and disease risk to wild fish populations (*Klinger & Naylor, 2012*). Environmental harm from the development of mariculture risks exacerbating some of the same human health factors it would seek to alleviate (*Cole et al., 2009*). Best practices for mariculture development are rapidly being developed and refined, and should be incorporated as an additional consideration in any nutrition-focused mariculture development (*Klinger & Naylor, 2012*).

Choosing appropriate mariculture species and practices will have significant implications not only for environmental impacts but also for nutritional quality and accessibility. Seafood products vary greatly in their specific nutrient content and therefore their potential contributions to human nutritional needs (*Glencross, 2009*; *Hixson, 2014*). A case study in Indonesia demonstrates the potential impact of farming a particularly vitamin A-rich species on local vitamin A deficiencies (*Fiedler et al., 2016*). Husbandry practices also have major impacts on nutritional quality, as well as environmental impacts. Species-specific feed compositions that provide digestible and appropriately-balanced nutrient compositions can improve fish growth and nutritional value while also minimizing waste due to nutrient indigestibility or oversaturation (*National Research Council, 2011*; *Hixson, 2014*). Because fish feed is the primary contributor to aquaculture's negative environmental impacts, improved practices that minimize overfeeding are critical in mitigating these deleterious effects (*Boyd et al., 2007*). Choosing local species can reduce the risk of invasive escapes as well as the spread of disease (*Diana et al., 2013*). Husbandry improvements such as reduced stock densities and appropriate water circulation can also reduce disease outbreaks and lessen the need for antibiotics, improving product quality for human consumers and

alleviating environmental issues with antimicrobial resistance (*Heuer et al., 2008*; *Diana et al., 2013*). As introduced earlier, the species produced in a given mariculture initiative can also determine seafood accessibility, with the nutritional and economic benefits of high-value species often failing to reach in-need populations (*Golden et al., 2017*).

The ability of a country to develop nutritionally-sensitive mariculture production will be extremely dependent on national policy and governance. *Thilsted et al. (2016)* advocate for 'nutrition-sensitive' fisheries and aquaculture policy that prioritizes context-specific nutritional needs and preferences. Policy incentivizing production of locally consumed, affordable, and nutritious products will be needed to prevent the dominant trajectory of export-oriented mariculture. Education and accessibility programs will also need policy support. Unfortunately, the overall mariculture opportunity score in our analysis is significantly correlated with the World Governance Indicator for political stability from the World Bank (http://info.worldbank.org/governance/wgi). This relationship means that countries needing nutritionally sensitive mariculture the most are also those with potentially the least capacity for implementation. This issue, however, is also an opportunity. The substantial overlap between economic and nutritional scores in our analysis suggests that in many countries, well-designed mariculture development programs and policies should be able to tackle both poverty alleviation and nutrition improvement outcomes.

There was not a significant correlation between nutritional and economic opportunity scores in our analysis, meaning that countries with high nutritional opportunity may not necessarily be places that can (or should) address these nutritional needs through further mariculture development. Instead, nutritional improvement in countries with highly developed mariculture industries may face more of a distributional rather than a production challenge (*Asche et al., 2015a*; *Watson et al., 2016*). Indonesia, for example, has a high nutritional need (nutritional opportunity score 0.86), while being the world's second-largest aquaculture producer. Strategies to better link mariculture and local nutrition in countries like Indonesia should consider existing mariculture industries and take advantage of them to the extent possible. Potential approaches include the transition of existing infrastructure or sharing processing facilities. Policy and market-based incentives would be critical in incentivizing a partial shift from export-oriented products to more accessible, low value species that could be sold and consumed domestically. Our analysis identifies the countries where this shift may be beneficial.

Finally, it is important to note that our analysis does not capture all of the nuances associated with the prospects and feasibility of mariculture development across the world. Institutional settings and market innovations may play a primary role when it comes to the actual development of technology and aquaculture (*Asche & Smith, 2018*). Further analysis and understanding of local and national-level institutions will be critical for effective mariculture development. Additionally, our global analysis, by necessity, is based upon data aggregated and averaged at the country level and may miss within-country as well as over-time dynamics. For example, just because a nation scores low on our nutritional opportunity scale does not necessarily mean there are not vulnerable segments of the population. South Africa, for example, receives a nutritional opportunity score of just 0.24. But it is also a nation of extreme inequality: The bottom 20% of the population receives less

than 5% of national income, while the top 20% receives more than 60% (*Statistics South Africa, 2014*). In rural areas, this inequality manifests in a 24.5% rate of youth stunting, much higher than the nation as a whole. Thus, in a nation like South Africa, there may be an opportunity for mariculture to contribute to a nutritional need, even if the nation has a (relatively) low nutritional opportunity score.

## CONCLUSION

By identifying important regional patterns in mariculture opportunity across three key combined measures of nutritional opportunity, economic opportunity, and seafood reliance, our analysis frames and focuses the necessary discussion on mariculture development and nutrition. An important finding is that nutritional and economic opportunities overlap in many nations, but come with significant challenges. As mariculture industries develop around the world, management choices will need to be made that balance high-value versus widely affordable species and promote nutrition-focused production expansion through appropriate public policy. Our analysis highlights the places where these policies could be impactful in promoting dual economic and nutritional goals, but further studies are needed on how to effectively capture the opportunities we have identified. What is the appropriate balance for a mariculture development program between production for local consumption versus high-value intensive production for export? How should mariculture species be prioritized for production, given country-specific conditions and nutritional needs? What are the environmental and ecological tradeoffs inherent in mariculture development, and how might development be guided to avoid extensive environmental degradation, further imperiling at-risk coastal populations? These questions are outside of the scope of this study, but remain essential topics for future research, as they will likely become key guiding principles to further mariculture development worldwide.

Mariculture production continues to expand and develop globally, and is a promising avenue to meet growing global nutritional challenges. Looking forward, if a development goal is to jointly develop mariculture and improve nutrition, countries with a higher reliance on seafood should be prioritized. In our analysis, countries with a high overall mariculture opportunity score not only have the economic development opportunity and a demonstrated nutritional need, but also the dietary preferences to link the two opportunities. Countries with high relative scores across the three components present potential win-win scenarios—where investing in nutrition and mariculture could have synergistic positive effects. Yet significant policy and institutional barriers remain in bridging the current gap between mariculture development and nutritional improvement. Addressing these barriers to achieve the development goal of improved global nutrition requires careful consideration, else we risk wasting a potentially powerful synergy between mariculture and nutrition opportunity.

## ACKNOWLEDGEMENTS

The authors would like to thank Bill Kuni and his partner Mary for their generous support of and comments on our work. Additionally, the early stages of this project benefitted

immensely from the contributions of Patricia Faúndez-Báez and the helpful comments of many other researchers from the Bren School and the National Center for Ecological Analysis and Synthesis. Finally, the authors would like to thanks two anonymous reviewers for constructive feedback.

### Funding
This work was supported by an H. William Kuni Research Award through the Bren School of Environmental Science and Management. The funders had no role in study design, data collection and analysis, decision to publish, or preparation of the manuscript.

### Grant Disclosures
The following grant information was disclosed by the authors:
Bren School of Environmental Science and Management.

### Competing Interests
The authors declare there are no competing interests.

### Author Contributions
- Owen R. Liu, Renato Molina and Margaret Wilson conceived and designed the experiments, performed the experiments, analyzed the data, prepared figures and/or tables, authored or reviewed drafts of the paper, approved the final draft.
- Benjamin S. Halpern conceived and designed the experiments, authored or reviewed drafts of the paper, approved the final draft.

### Data Availability
  The raw data are provided as a Supplemental File.

### Supplemental Information
Supplemental information for this article can be found online at http://dx.doi.org/10.7717/peerj.4733#supplemental-information.

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
