# Peer review of "Global opportunities for mariculture development to promote human nutrition"

_PeerJ, doi:10.7717/peerj.4733_

## Round 0.1 · original submission · Minor Revisions

The reviewers have suggested a number of relatively minor revisions which you need to respond to.

Reviewer 1 ·

Basic reporting

This paper is well-written and provides really interesting food for thought regarding the potential for expansion of mariculture activities in alignment with appropriate policies to address nutritional deficiencies. The authors are clear about the limitations of such a broad approach and its individual components as proxies (e.g. ratio of aquaculture production to fisheries production as a proxy for infrastructure). There are lots of insightful and interesting parts throughout the discussion and conclusion. The structure is clear and the data are presented effectively.

Minor comments:
- Convert Final opportunity score rank table from landscape to portrait
- Add accents to Bene throughout the manuscript (sometimes you do, sometimes you don't)
- Lines 307-313 - you switch back and forth between "aquaculture" and "mariculture" here; is this intentional?
- I'd delete the first sentence of this paragraph (Line 361-362) and just lead with the second, which is more important and stronger.

Experimental design

The authors are clear about their methodology, why some countries needed to be excluded due to data gaps, and the risks of aggregation (e.g. the loss of local / regional diversity within countries). The sensitivity check on variables is good, and the transparency resulting from the sharing of all data is helpful. See above for concerns about including all coastal countries in this analysis and conflating reliance/preference, etc. The proxy for infrastructure (capture fisheries / aquaculture production) also becomes problematic in countries with small production volumes, as FAO data is most accurate for large-volume fisheries and aquaculture production and becomes shakier at smaller volumes (i.e. a small island state may have very precise tonnage reports for tuna fisheries, but far less accurate reports for smaller or less well-monitored fisheries, especially artisanal and subsistence fisheries), yet the tuna catch would likely be exported and have little nutritional contribution to local communities, while the artisanal/subsistence fisheries would be largely missed, while being more important in this regard.

In short, the authors have made it clear that this is a big picture, rough assessment, and I believe it has value as such. But perhaps a bit more attention to the data limitations and their implications for the different components of the index would be helpful.

Validity of the findings

The inclusion of highly industrialized countries (where populations have access to diverse foods rich in micronutrients) and developing countries (where access to such foods is more mixed) within the same index caused a number of problems throughout the paper. The definition, for instance, of "reliance" on seafood for this entire suite of countries is problematic, as it certainly become "preference" rather than reliance for populations with access to diverse foodstuffs and the means to purchase it. Highly industrialized countries that consume lots of seafood, but which have access to any number of other foods that could provide this nutrition are unlikely to face micronutrient deficiencies related to fluctuations in fish availability or price. Likewise for nutritional opportunity, according to the index, countries like Venezuela and Syria are among those with the least nutritional opportunity, ranking below countries like Denmark and Canada. More clarity may be possible by simply dividing the index into several categories of countries (e.g. based on income levels), or in your graphics by using coloring similar to Golden et al. (2016) that draws emphasis away from countries with little reliance on seafood for nutrition. Or perhaps just focus on coastal, low-income food deficit countries?

Another issue that was missing was a greater focus on the actual mariculture species being produced (you hint at this on lines 337-338). It is useful that the authors highlight a divide between high volume and high value production, and also the disconnect between mariculture production levels and nutritional outcomes. Yet I wonder about the argument for including high value mariculture species in this analysis. Residents in low-income food deficit countries are unlikely to be able to afford high value mariculture outputs. Adding further information to the introduction or discussion on which mariculture species are the best candidates (i.e. low cost, high nutritional value, generalist, less susceptible to disease, etc) for reducing food insecurity and nutritional deficiencies would be helpful to frame this issue.

Additional comments

This is really interesting and exciting work! Good luck with it! While having a global index with lots of countries may feel nice, I have a hunch that limiting your index to LIFDCs or least developed countries, etc may help to clarify some of your arguments and conclusions, and feed into clearer policy recommendations. Good luck!!

Reviewer 2 ·

Basic reporting

Very good

Experimental design

Very good

Validity of the findings

Very good

Additional comments

This is a really useful study using a novel approach to show where mariculture has the largest potential to influence human nutrition.
Ln. 62. Please note that the per capita consumption of 20.3 is the highest aver recorded.
And while I agree that nutritional importance of fish is part of the explanation, I think you should add somthing to the effect that the main driver has been innovatins that has reduced production cost and increased competitiveness not only towards marine fish, but also other foods (Smith et al,, 2010 Science; Kumar and Engle, 2016, Reviews in Fisheries and Aquaculture).
Ln 76. While this is true for many species, it it should be nuanced as e.g. Belton et al (2018) show that a significant portion of aquaculture has made seafood more available in countries like China, Bangladesh, Indonesia and Egypt. Still this do not affect you main question about mariculture, as the main species here are freshwater species. However, it do suggest that mari-culture can become important.
Ln 156. I realize that all metrics has to be somwhat pragmatic given the data availability, but I think you should also acknowledge weaknesses. For instance, cultural preference is also an expression of tradiotinal availability, and there are numerous examples of how eating habits change in surprising directions with economic opportunity with Japan´s declining seafood consumption and increasing meat consumtion as an example. I think your economic metrics are a bit more problematic than your other metrics as they are even more grounded in what has been, and not in what is possible.
Ln. 235. I assume that also exludes nations with significant fisheries in large lages like Uganda? Please note.
Ln 262. This may not be too surprising since aquaculture does not happen to a large extent in developed countries.
Ln. 266. Latin America is really interesting since that is also the region where the relative growth is predicted to be fastes in most general aquaculture production forecasts such as the WorldBand FAO Fish to 2030 project (Kobayashi et al 2015, Aquaculture Economics and Management).
Ln 376. I think that you somewhere need to state cleare that there are also other reasons for why aquaculture production may grow in a country such a natural endowment etc., that may lead aquaculture to increase also places that your model does not indicate. Asche and Smith (2018, Food Policy) is a recent example of a study that argue that increased aquaculture production is largely a function of incentives. Howevert his may also increase availability for nutiritional poor, as seafood to developing cointries are increasing quite rapidly. For instance, Nigeria is now the world´s 11th largest seafood imorter by quantity.

---

## Round 0.2 · accepted · Accept

Great job with the revisions. You addressed all the reviewer suggestions and the paper is now accepted

#